# Characterizing physical activity bouts in people with stroke with different ambulation statuses

David Moulaee Conradsson[1,2]*, Burcin Aktar[3,4], Lucian Bezuidenhout[1,5]

1 Department of Neurobiology, Care Sciences and Society, Division of Physiotherapy, Karolinska Institutet, Stockholm, Sweden, 2 Women's Health and Allied Health Professionals Theme, Medical Unit Allied Health Professionals, Karolinska University Hospital, Stockholm, Sweden, 3 Institute of Health Sciences, Dokuz Eylul University, Izmir, Turkey, 4 Faculty of Physical Therapy and Rehabilitation, Dokuz Eylul University, Izmir, Turkey, 5 Department of Health and Rehabilitation Sciences, Division of Physiotherapy, Stellenbosch University, Cape Town, South Africa

* david.m.conradsson@ki.se

## Abstract

### Background and Purpose

While physical activity is crucial for maintaining function, health, and well-being after a stroke, there is limited understanding of how individuals post-stroke accumulate their daily activity in terms of bouts and intensities. This study aimed to characterize and contrast the daily patterns, frequency and intensity of physical activity bouts between people post stroke with different ambulation statuses compared to healthy controls.

### Methods and Materials

In this cross-sectional study, physical activity bouts patterns, frequencies, and intensities were evaluated using Actigraph GT3X+ accelerometers across three groups: 17 limited community ambulators (LCA) post-stroke (walking speed: < 0.8 m/s), 22 community ambulators (CA) post-stroke (walking speed: ≥ 0.8 m/s), and 28 healthy controls.

### Results

People post stroke primarily engaged in 1–5 min bouts (LCA: 79%, CA: 76%), with less frequent engagement in 5–10 min (12–14%) and > 10 min bouts (9–10%) during the day. The LCA group engaged comparable or greater time spent in light physical activity during >5–10 and > 10 min bouts compared to CA and healthy controls, but less time in moderate to vigorous physical activity (P < .009). Both post-stroke ambulation groups were most active between 12–5 pm.

### Conclusions

CA people post stroke exhibited patterns similar to healthy controls in physical activity bouts, whereas LCA primarily engaged in short bouts and light activity. In the context

**Data availability statement:** Since data can indirectly be traced back to the study participants, according to the Swedish and EU personal data sharing legislation, access can only be granted upon request from the Research Data Office at Karolinska Institute (rdo@ki.se). Any sharing of data will be regulated via a data transfer and use agreement with the recipient and require ethical approval from the Regional Board of Ethics in Stockholm.

**Funding:** This study was supported by the Norrbacka-Eugenia foundation, Promobilia foundation, and NEURO Sweden.

**Competing interests:** The authors report there are no competing interests to declare.

of secondary stroke prevention, encouraging LCA people post stroke to engage in frequent short bouts of moderate to vigorous physical activity or longer bouts of light physical activity might be realistic targets to improve cardiovascular health.

---

## Introduction

Physical activity (PA) is essential after stroke to sustain function, health, and well-being [1–3] and reduce the risk of recurrent stroke [4]. In particular, engaging in regular PA after a mild stroke have shown to reduce the 5-year risk of stroke-related disability by 44% and stroke recurrence by 48% [4]. Still, people after stroke compared to healthy individuals spend more time sedentary (10.9 vs 8.2 hours) [5–7] and take fewer steps per day (4078 vs 8338 steps) [6]. Breaking up sedentary time into short bouts of PA has been shown to improve recurrent stroke risk factors; such as systolic blood pressure [8,9] and variability in glucose levels [10]. However, previous studies have mainly investigated the total PA per day in people after stroke (e.g., number of steps per day) [7,11] with few studies determining how PA is accumulated with different bout durations and diurnal patterns (i.e., the time of day when most and least PA occurs) [12–14]. Knowledge of the diurnal patterns, frequency, and intensity of PA bouts could inform future strategies for PA promotion following stroke.

Mobility status and gait speed significantly influence the ability of individuals post-stroke to engage in PA [15]. Self-selected gait speed is commonly used as a proxy measure to assess community ambulation post-stroke, with individuals classified as limited community ambulators (LCA) if their gait speed is ≤ 0.8 m/s and as community ambulators (CA) if their gait speed is ≥ 0.8 m/s [16]. Community ambulators post-stroke are often capable of independent mobility outside the home environment, including confidently navigating uneven terrain and other public spaces [17]. In contrast, LCAs are often limited to the home or its surroundings and are more vulnerable to secondary health complications resulting from sedentary behavior [16].

The composition of PA behavior, i.e., light intensity physical activity (LIPA) and moderate to vigorous physical activity (MVPA), is important for cardiovascular health post stroke [18]. Light intensity physical actvity and MVPA are movement behaviors corresponding to energy expenditures of 1.0–3.0 Metabolic Equivalent of Tasks (e.g., slow walking) [19] and ≥ 3.0 Metabolic Equivalent of Tasks (e.g., brisk walking and jogging) [20], respectively. For example, focusing on accumulating short bouts of MVPA lasting 1–5 minutes could be a realistic goal for individuals post stroke, particularly for those facing challenges related to old age and walking limitations [21]. Furthermore, Hassett et al. showed that individuals post stroke engaged in a similar frequency of 1–5 min PA bouts as age-matched healthy controls yet participated in fewer bouts lasting over 30 minutes [12].

Previous studies determining the diurnal pattern of PA have shown that the proportion of each hour spent sedentary increases across the day in people post stroke [14,22,23]. Joseph et al., (2020) showed that the diurnal PA pattern was related to ambulation status post stroke. Specifically, LCA people post stroke demonstrated a

constant diurnal pattern with low levels of PA without any variation, whereas CA people post stroke were most active in the morning followed by a gradual reduction in PA throughout the day [24]. Furthermore, Tieges et al., 2015 investigated the diurnal sedentary time curves in people post stroke and showed decreasing sedentary time in the morning and gradually increasing sedentary time in the afternoon and evening [22,24].

To our knowledge, no previous study has analyzed the PA patterns in relation to different PA bout durations and compared the patterns of people post stroke with healthy individuals. This is important given that people post stroke are often older [21] and the decrease in PA with age [25]. Moreover, most studies investigating PA behavior in individuals post stroke focus on those with mild ambulation impairments [24]. Exploring PA behavior in individuals post stroke with diverse ambulation statutes could provide a better foundation for the development of tailored interventions for PA promotion after stroke. The aim of this study was therefore to characterize and contrast the diurnal patterns, frequency and intensity of PA bouts between LCA and CA people post stroke compared to healthy controls.

## Materials and methods

### Study setting and ethical considerations

This cross-sectional study was conducted in accordance with the Declaration of Helsinki principles and was approved by the Regional Board of Ethics in Stockholm (2017/1626–31 and 2018/2524–32). All study participants provided written informed consent before their enrollment in the study.

### Study participants

People post stroke were recruited through a rehabilitation clinic. Inclusion criteria were clinical diagnosis of stroke ≥ 3 months before study enrollment and being able to walk indoors with/without a walking aid. Exclusion criteria were cognitive impairments, severe neglect and global aphasia affecting the ability to provide written informed consent. Cognitive impairments were screened using the Montreal Cognitive Assessment (MoCA) [26], neglect was assessed using the Star Cancellation Test [27], and the ability to communicate through speech was evaluated informally during a telephone interview and while completing the MoCA. Healthy individuals with no medical condition that restricted their ability to participate in PA were recruited through advertisements and senior organizations and served as a reference group.

### Data collection

Data collection was performed on one occasion at a rehabilitation clinic by a physiotherapist with experience in stroke rehabilitation. This process included standardized interviews of demographics (age, sex, height, and weight), clinical assessments and performance-based clinical tests, and assessment of PA in daily life. For the healthy control group, data collection was limited to demographics and assessment of PA using accelerometers.

### Clinical assessments and performance-based clinical test

Physical impairment post stroke was measured using the arm, hand, leg, foot and postural control sub-domains of the *Chedoke-McMaster Stroke Assessment* (CMSA) [28,29]. The functional level for each domain was assessed on a 7-point Likert scale ranging from 1 (no recovery) to 7 (complete recovery). The CMSA was developed for assessment of people with stroke and shown to be valid measure of physical impairment post stroke [30]. For the upper limb domain, we analyzed a sum score of the arm and hand subsections and for the lower limb domain, we used a sum score of the leg and foot subsections (i.e., a total score of 14). The *Fatigue Severity Scale*, a valid and reliable measure for people post stroke [31], was used to evaluate the severity of fatigue, which is composed of 9-items [32], and is scored using a 7-point Likert-type scale (1 = strongly disagree, 7 = strongly agree) which is summarized as an average score. The anxiety and depression subdomains of the *Hospital Anxiety and Depression Scale* (HADS) were used to assess anxiety and depressive

symptoms [33]. The HADS has been shown to be a suitable instuments for assessing anxiety and depression syptoms in people post stroke [34]. Each subdomain consists of 7 items graded on a 4-point (0–3) scale which is presented as a sum score (max score 21, higher scores indicating greater anxiety or depression symptoms). The MoCA, a reliable tool for people post stroke [35], was used to assess cognitive functions, i.e., short-term memory recall tasks, visuospatial abilities, executive function, attention, concentration, and working memory, language, which is summarized as a sum score (range 0–30, higher score = greater cognitive function) [26].

Walking ability was assessed using walking speed (m/s) during a modified Six Minutes Walk Test [36]. Participants were instructed to "walk at their normal comfortable pace for 6 min" on a track with 180-degree turns every 60 meters. Average walking speed (meter/second) was used to classify people post stroke as limited community ambulators (LCA, walking speed: < 0.8 m/s) or community ambulators (CA, walking speed: ≥ 0.8 m/s) [24,37].

## Assessment of physical activity

Physical activity was measured using a three-axial accelerometer (Actigraph GT3X+) with a sampling rate of 30 Hz [38,39]. The participants were instructed to wear the accelerometer around their waist; people post stroke on the non-paretic side and healthy individuals on the dominant side, for 7 consecutive days during waking hours. A 7 day period captures the variability in the activity pattern (i.e., weekdays vs. weekends) and provides a reliable estimate of habitual activity [40]. Participants were asked to record the device wear time in a daily diary and only take the device off while showering or bathing [41].

The raw PA data were downloaded and converted to counts using ActiLife 6 (version 6.13.4) software, where a count is defined as the acceleration signal crossing a proprietary amplitude cut point [42]. Furthermore, the PA data was processed into 1-minute epochs and divided into daily segments between 6 am and 10 pm. Subsequently, a daily non-wear time algorithm, in accordance with Choi et al. was applied [43]. Briefly, the algorithm examines 90 min of continuous zero counts and 2 min of non-zero counts while considering 30 min of consecutive zero counts before and after the detected non-zero counts [43]. The periods of wear time were confirmed by the participant's diaries and days with ≥ 10 hours of PA data were used for analysis [42].

To analyze PA bouts, one-minute epochs with < 100 vertical counts/min were classified as sedentary and epochs with ≥100 vertical counts/min were classified as PA [44,45]. The 100 counts/min is a robust marker for distinguishing sedentary and PA across different study populations [46–48]. PA bouts of different continuous durations of 1–5 min, > 5–10 min, and > 10 min between 6 am and 10 pm were detected. Thereafter, the median time spent in LIPA (100–1041 vertical counts/min) and MVPA (≥ 1041 vertical counts/min) [49] were determined for each bout duration [46–48]. To classify the temporal PA pattern, the mean number of continuous PA bouts for lengths between 1–5 min, > 5–10 min, and > 10 min for each hour between 6–12 am (morning), 12–5 pm (afternoon) and 5–10 pm (evening) were calculated.

## Statistical analysis

Data were analyzed using IBM SPSS Statistics (Version 24.0). Demographics, stroke impairments, fatigue, anxiety and depressive symptoms, cognition and walking ability were reported using numbers and percentages (%) for categorical data, while median and interquartile ranges (IQR) were used for continuous data. These variables were compared between CA and LCA people post stroke using the chi-square, Fisher's exact test and Mann–Whitney U-test.

The mean number of 1–5 min, > 5–10 min, and > 10 min PA bouts and mean time spent in LIPA and MVPA did not follow a normal distribution, therefore, the Kruskal-Wallis test was used to compare these variables between CA and LCA people post stroke and healthy controls. The post hoc analysis was performed using the Mann-Whitney U tests with Bonferroni corrections, with an adjusted P-value < 0.017 considered statistically significant.

# Results

## Participants' characteristics

Out of the 45 participants post stroke, 6 participants were excluded due to not having sufficient PA data. Thus, the study included 39 people post stroke (LCA: n = 17, CA: n = 22) alongside 28 healthy controls. The characteristics of the study participants are detailed in Table 1. Limited community ambulators post stroke had greater impairments regarding upper limb, lower limb, postural control and walking speed compared with the CA group ($P < 0.004$, Table 1). Median age and sex distribution were similar among the healthy control group (65.5 years; IQR: 57.5–73.0, 32% females) and the LCA (68.3 years; IQR: 59.1–72.5, 35% females) and CA people post stroke (62.7 years; IQR: 53.1–73.6, 45% females).

## Physical activity bouts

Table 2 shows the number of daily PA bouts and their corresponding intensities among people post stroke and the healthy control group. Limited community ambulators post stroke engaged in fewer 1–5 min bouts per day (62.2) compared to CA people post stroke (71.7, $p = 0.012$) and the healthy control group (72.1, $p = 0.011$). The LCA engaged in significantly less number of > 5–10 min bouts ($p < 0.013$) and > 10 min bouts ($p < 0.017$) compared to CA and the health control group. The LCA group participated in significantly more LIPA ($p \leq 0.005$) in > 5–10 min bouts and significantly less MVPA in > 5–10 min ($p \leq 0.011$) and > 10 min ($p \leq 0.006$) bouts compared to the CA and the healthy control group.

## Physical activity pattern over the day

The results showed varying patterns in the frequency of PA bout lengths across the study groups (Fig 1). The LCA group demonstrated an increase in the number of PA bouts from the morning to the afternoon for both 1–5 min ($p < .005$) and > 5–10 min bouts ($p < 0.020$), while no significant changes were observed throughout the day for > 10 min bouts ($p > 0.245$). For the CA group, while no changes occurred for 1–5 min bouts ($p > .0082$), there was a significant increase in the number of bouts from morning to afternoon for both >5–10 min ($p < 0.003$) and > 10 min bouts ($p < 0.006$). The daily pattern in PA bouts in the healthy control group largely resembled those observed among CA people post stroke.

**Table 1. Participants' characteristics of people post stroke. Data is presented as median and interquartile range.**

|  | Limited community ambulators (n = 17) | Community ambulators (n = 22) | *P* |
|---|---|---|---|
| Age (years) | 68.3 (59.1–72.5) | 62.7 (53.1–73.6) | 0.475 |
| Female sex, n (%) | 6 (35) | 10 (45) | 0.522 |
| BMI (kg/m²) | 25.7 (22.5–28.6) | 24.8 (23.5–29.4) | 0.889 |
| Time since stroke (years) | 2.0 (0.6–3.3) | 1.0 (0.5–2.1) | 0.257 |
| CMSA upper limb | 12.0 (9.5–14.5) | 17.0 (14.7–20.0) | <0.001 |
| CMSA lower limb | 10.0 (9.0–11.0) | 11.0 (10.7–13.2) | 0.004 |
| CMSA postural control | 3.0 (2.0–5.0) | 6.0 (5.0–7.0) | 0.001 |
| Use of walking aid, n (%) | 14 (82) | 9 (41) | 0.009 |
| Self-selected walking speed (m/s) | 0.4 (0.3–0.6) | 1.0 (0.9–1.2) | <0.001 |
| Fatigue Severity Scale | 3.2 (2.3–5.1) | 4.1 (2.5–5.0) | 0.605 |
| HADS anxiety | 2.0 (0.5–4.0) | 2.0 (1.0–4.0) | 0.823 |
| HADS depression | 2.0 (1.0–5.5) | 3.0 (1.0–4.5) | 0.823 |
| Montreal Cognitive Assessment | 26.0 (21.0–27.5) | 25.0 (22.0–27.2) | 0.808 |

Abbreviations: BMI, body mass index; CMSA, Chedoke-McMaster Stroke Assessment; HADS, Hospital Anxiety and Depression Scale

**Table 2. Comparison of physical activity bouts between limited community ambulators and community ambulators post stroke, and healthy controls. Data are presented as median and interquartile range.**

| | Groups | | | P-value post hoc analysis** | | | |
|---|---|---|---|---|---|---|---|
| | Limited community ambulators (n = 17) | Community ambulators (n = 22) | Healthy controls (n = 28) | Kruskal-Wallis test* | Limited community ambulators vs community ambulators | Limited community ambulators vs healthy controls | Community ambulators vs healthy controls |
| **1-5 min bouts** | | | | | | | |
| Number of bouts | 62.2 (47.1–72.1) | 71.7 (66.7–79.9) | 69.7 (66.5–81.6) | **0.018** | **0.012** | **0.011** | 0.815 |
| LIPA (min) | 1.6 (1.5–1.7) | 1.8 (1.6–1.8) | 1.9 (1.8–1.9) | **<0.001** | 0.018 | **<0.001** | 0.022 |
| MVPA (min) | 0.3 (0.2–0.6) | 0.4 (0.3–0.5) | 0.5 (0.4–0.6) | 0.112 | 0.255 | 0.052 | 0.233 |
| **>5–10 min bouts** | | | | | | | |
| Number of bouts | 9.5 (6.2–12.8) | 13.1 (10.2–16.2) | 17.0 (15.9–19.3) | **<0.001** | **0.013** | **<0.001** | **0.001** |
| LIPA (min) | 6.4 (6.0–6.9) | 5.4 (4.5–6.2) | 5.7 (5.4–6.1) | **0.003** | **0.005** | **0.002** | 0.538 |
| MVPA (min) | 1.3 (0.8–2.5) | 2.6 (1.9–3.5) | 2.4 (1.8–2.9) | **0.004** | **0.002** | **0.011** | 0.174 |
| **>10 min bouts** | | | | | | | |
| Number of bouts | 7.0 (1.5–10.3) | 9.3 (7.5–13.4) | 15.4 (12.7–17.7) | **<0.001** | 0.017 | **<0.001** | **<0.001** |
| LIPA (min) | 9.9 (5.8–11.7) | 7.6 (6.7–9.0) | 9.1 (7.4–11.1) | 0.067 | 0.067 | 0.725 | 0.033 |
| MVPA (min) | 2.0 (0–8.6) | 7.1 (5.7–9.5) | 9.1 (5.8–10.8) | **0.004** | **0.006** | **0.002** | 0.319 |

Abbreviations: LIPA, light intensity physical activity; MVPA, moderate-to-vigorous intensity physical activity

* $P < 0.05$, Kruskal-Wallis test

**$P < 0.017$, Post hoc (Mann-Whitney U test) analysis with Bonferroni corrections

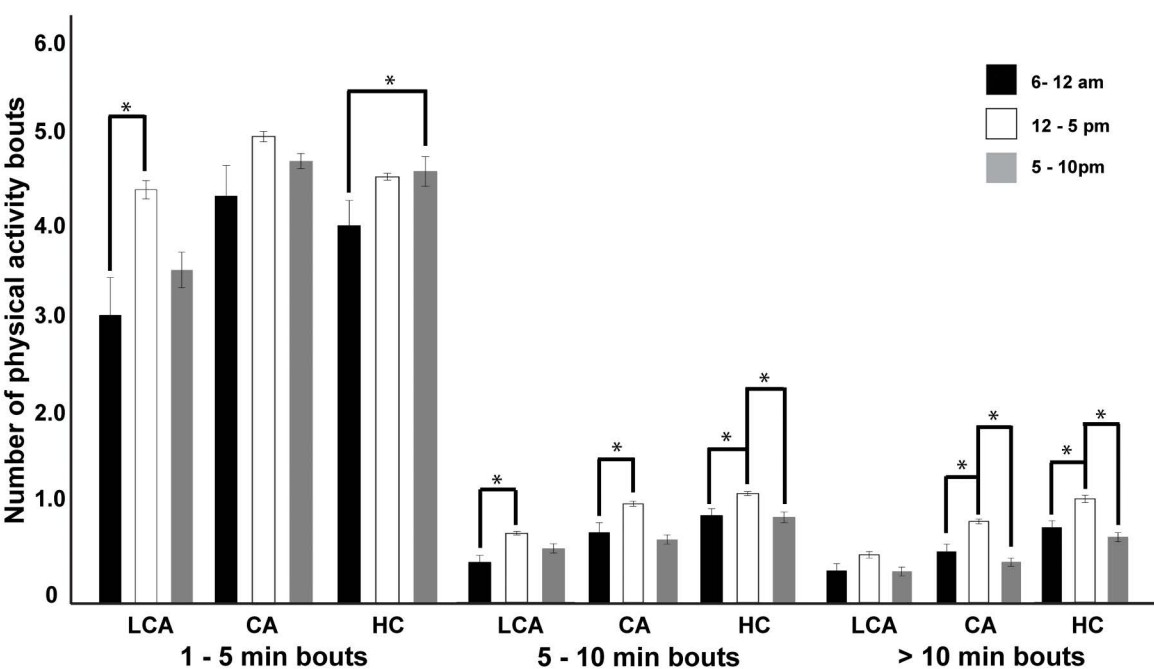

**Fig 1. The mean daily number of physical activity bouts categorized by duration (1-5 min, > 5-10 min, and > 10 min) during morning (6-12 am), afternoon (12-5 pm), and evening (5-10 pm) periods for limited community ambulators (LCA) and community ambulators (CA) post-stroke, and healthy controls.** * Denotes the statistically significant results.

## Discussion

Few studies have determined the diurnal pattern, frequency and intensity of PA among people post stroke with varying ambulation status. Our key findings revealed that CA people post stroke demonstrated PA bouts characteristics resembling those among the healthy control group, while LCA primarily engaged in short bouts and LIPA. Tailored interventions corresponding to post stroke ambulation status are important to promote health-enhancing PA.

Similar to our findings, Hassett et al. [12] found that community-dwelling individuals post stroke with an average gait speed of 0.8 m/s accumulated the majority (64%) of their PA in < 5 min bouts. However, in contrast to our results, this study also found similar engagement in short PA bouts among healthy controls. In the study by Hassett et al [12], both stroke and healthy participants were approximately 5 years older compared to participants in our study. This age difference might explain the varying results, as studies commonly link a decrease in PA with advancing age [25]. Roos et al. [50] also found that people post stroke and healthy controls accumulated all their walking bouts at short durations (0–40 steps) corresponding to the 1–5 min bouts in the present study [50]. It is worth noting the methodological differences in the assessment of PA, i.e., the definition of PA bouts, placement and type of accelerometer used, in the aforementioned studies. For instance, Hassett et al. [12] defined PA bouts as time spent standing, walking, or transitioning from sitting to standing using chest, thigh, and foot-worn accelerometers, whereas Roos et al. [50] defined PA bouts as walking using ankle-worn accelerometers. These discrepancies introduce inconsistencies in classifying and quantifying PA bouts and could lead to variations in reported PA behaviors, ultimately affecting the overall comparability of the studies.

The present results showed that LCA people post stroke engaged in similar or greater time in LIPA but less time in MVPA during bouts >5 min compared to CA people post stroke and healthy controls. This result may be explained by reduced functional capacity among LCA compared to CA people post stroke, which may restrict the capacity to sustain MVPA for longer bouts. In line with this, it was more common to use a walking aid among LCA (82%) than the CA group (41%). Consequently, due to the impairments in postural control and lower limb function observed in the LCA group, longer bouts of LIPA may be more attainable for this group. It is important to highlight that low intensity activities of everyday living, such as slow walking, household tasks, or gardening, are typically classified as LIPA. However, for individuals with more pronounced disabilities, these activities may require greater effort and energy expenditure, potentially reaching MVPA levels [51]. Regardless of this uncertainty, engagement in LIPA is associated with the recovery of gait in the subacute phase post stroke [52] and reduced risk for premature mortality in the general population [53]. The present results are therefore encouraging, especially for LCA people post stroke who often are unable to perform longer bouts of MVPA. Furthermore, our results regarding the daily patterns of PA are in line with previous studies exploring daily PA in people post stroke [22,24] and people with Parkinson's disease [54]. Previous research has demonstrated that healthy adults increase their PA in the morning and afternoon and spend more time sedentary in the evening [54,55].

To date, there is conflicting evidence on whether continuous PA of longer bouts is more beneficial to cardiovascular health than multiple short PA bouts, with increasing evidence suggesting cardiorespiratory fitness may be achieved through accumulating LIPA and MVPA of different bout durations [56]. Our findings underscore the significance of directing PA promotion efforts toward LCA people post stroke. This could involve integrating exercise with behavior change techniques to enhance both capacity and motivation for sustained PA engagement [57,58]. Still, prior systematic literature reviews indicate a notable gap in research, as individuals with restricted ambulation status are frequently overlooked in studies aimed at assessing tailored support for PA post stroke [59]. Addressing this gap is especially important since LCA individuals post stroke are particularly vulnerable to secondary health complications due to sedentary behavior [16]. Fulfillment of the recommendation of 150 min of MVPA per week might not be achievable for LCA people post stroke and instead, it could be more realistic to target longer bouts of LIPA to improve cardiovascular health.

The strength of this study was the use of accelerometers to characterize PA bouts and patterns among people post stroke with different ambulation statuses and healthy controls. Nevertheless, some methodological considerations require attention. Firstly, the sample size of our study was relatively small, and only independently ambulating people post stroke were

included. This limits the generalizability of the study results to people post stroke with greater walking limitations. Furthermore, the median age of the stroke participants in our study (64 years) was younger than the reported average for the population in Sweden (75 years) and the percentage of women (41%) was slightly lower [60]. Future studies should aim to include a more diverse sample, particularly with greater representation of women, older adults, and individuals with a wider range of impairments following a stroke. This will enhance the generalizability of the findings. Secondly, the MVPA cutoff utilized in our study was developed for healthy older adults [49], corresponding to a walking speed of 0.9 m/s. This is similar to the median self-selected walking speed in the CA post-stroke group in the present study (1.0 m/s) but notably higher than the gait speed in the limited ambulating group (0.4 m/s). Consequently, due to the increased energy expenditure during low-intensity activities of everyday living (i.e., walking) post-stroke [51], walking at a slower pace could potentially constitute MVPA for stroke survivors with more severe disability, suggesting that our results might underestimate the intensity of PA among LCA people post stroke. To enhance the accuracy of PA assessment in the stroke population, it is imperative to develop and validate PA intensity cut-points that accurately reflect the physiological capabilities of individuals with diverse ambulation statuses post stroke. Thirdly, our methodology differs from previous studies measuring PA bouts in people post stroke [12,50], making it difficult to compare the results across the different studies. Therefore, future research should investigate how accelerometer placement affects the accuracy of capturing various domains (i.e., steps, MVPA) of PA post stroke. Finally, the study design limits the ability to infer causality or observe changes in PA over time. Understanding these changes is crucial for evaluating the long-term impact of PA on stroke recovery and cardiovascular health. To gain deeper insights, future research should focus on conducting longitudinal studies to examine the levels and patterns of PA following a stroke.

## Conclusion

Community ambulating people post stroke demonstrated PA bouts characteristics resembling those among healthy controls. Limited community ambulating people post stroke engaged in similar or more LIPA as compared to CA and healthy controls, but less time in MVPA. In the context of secondary stroke prevention, encouraging LCA people post stroke to engage in frequent short bouts of MVPA or longer bouts of LIPA might be realistic targets to improve cardiovascular health.

## Acknowledgment

The authors thank all the participants for their valuable time and efforts.

## Author contributions

**Conceptualization:** David Moulaee Conradsson, Lucian Bezuidenhout.

**Data curation:** David Moulaee Conradsson.

**Formal analysis:** Burcin Aktar, Lucian Bezuidenhout.

**Funding acquisition:** David Moulaee Conradsson.

**Investigation:** David Moulaee Conradsson, Lucian Bezuidenhout.

**Methodology:** David Moulaee Conradsson, Burcin Aktar.

**Project administration:** David Moulaee Conradsson, Lucian Bezuidenhout.

**Resources:** David Moulaee Conradsson.

**Supervision:** David Moulaee Conradsson, Lucian Bezuidenhout.

**Validation:** Burcin Aktar.

**Writing – original draft:** David Moulaee Conradsson, Burcin Aktar, Lucian Bezuidenhout.

**Writing – review & editing:** David Moulaee Conradsson, Burcin Aktar, Lucian Bezuidenhout.

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
