## [Decision Letter · Decision Letter 0]

15 Aug 2024

PONE-D-24-27556

Characterizing physical activity bouts in people with stroke with different ambulation statuses

PLOS ONE

Dear Dr. Conradsson,

Thank you for submitting your manuscript to PLOS ONE. After careful consideration, we have decided that your manuscript does not meet our criteria for publication and must therefore be rejected.

I am sorry that we cannot be more positive on this occasion, but hope that you appreciate the reasons for this decision.

Kind regards,

Hidetaka Hamasaki

Academic Editor

PLOS ONE

Reviewers' comments:

Reviewer's Responses to Questions

**Comments to the Author**

1. Is the manuscript technically sound, and do the data support the conclusions?

Reviewer #1: Yes

Reviewer #2: Yes

2. Has the statistical analysis been performed appropriately and rigorously? 

Reviewer #1: Yes

Reviewer #2: Yes

3. Have the authors made all data underlying the findings in their manuscript fully available?

Reviewer #1: No

Reviewer #2: Yes

4. Is the manuscript presented in an intelligible fashion and written in standard English?

Reviewer #1: Yes

Reviewer #2: Yes

5. Review Comments to the Author

**Reviewer #1:**  The paper investigates how physical activity (PA) varies between stroke survivors with differing ambulatory capabilities and compares them with healthy controls. Using accelerometers, the study characterizes and contrasts daily PA patterns, frequencies, and intensities among limited community ambulators, community ambulators, and healthy controls. It has several strengths:

• The paper addresses an important gap in the literature by exploring the PA patterns in post-stroke individuals with varying ambulation statuses, providing a nuanced understanding of their activity profiles.

• The use of accelerometers (Actigraph GT3X+) for objective measurement of PA provides accurate and reliable data, enhancing the credibility of the findings.

• The paper provides clear and practical recommendations for improving cardiovascular health in stroke survivors, emphasizing tailored interventions based on ambulation status.

• Ethical considerations were well-addressed, including approval from the Regional Board of Ethics in Stockholm and informed consent from participants.

However, it has a few major weaknesses:

• Small Sample Size: The study only includes 39 participants post-stroke, limiting the generalisability of the findings to a broader population of stroke survivors.

• Limited Diversity in Participants: The study's participants are predominantly younger and have fewer women than the typical stroke population in Sweden, potentially affecting the applicability of the findings.

• MVPA Cutoff Limitations: The MVPA cutoff used was developed for healthy older adults and might underestimate the intensity of PA for those with more severe disabilities. This could lead to misclassification of activity intensity, especially for the LCA group.

• Lack of Longitudinal Perspective: The study is cross-sectional, limiting the ability to infer causality or observe changes in PA patterns over time.

• Methodological Differences: The paper mentions differences in accelerometer placement and PA bout definitions compared to previous studies, which may affect comparability.

• Data Sharing Restrictions: The data availability statement indicates restrictions due to privacy concerns, limiting access for further validation and research.

• Lack of Severe Cases: The exclusion of individuals with severe ambulation impairments limits the study’s applicability to all stroke survivors.

In addition, a few minor comments

1. Language and Clarity: Some sections could benefit from clearer language to enhance readability.

2. Figures and Tables: Ensure all figures and tables are clearly labeled and referenced in the text for better comprehension.

3. Data Presentation: More detailed presentation of the data analysis methods could improve transparency and reproducibility.

Overall, this paper is more suited for the audience in the specialist physiotherapy journal rather than PLOS ONE.

**Reviewer #2: ** The results presented in the current report require a more thorough and detailed analysis to fully understand their implications. It is essential to provide a comprehensive explanation of what the values signify in the context of the study. This includes interpreting the numerical data and explaining how these values relate to the objectives and hypotheses of the research.

A detailed breakdown should include:

Contextual Interpretation: Elaborate on how the values fit within the broader context of the study. This means discussing the relevance of these results to the research questions and how they align with or diverge from expected outcomes.

Statistical Significance: Clarify the statistical significance of the results. Provide information on the confidence intervals, p-values, or other statistical metrics used to determine the reliability of the findings.

Comparative Analysis: Compare the results with previous studies or benchmarks. Explain how these values contribute to or challenge existing knowledge in the field.

Practical Implications: Discuss the practical implications of the results. What do these values mean for practitioners, stakeholders, or policymakers? How can they be applied in real-world scenarios?

Visual Aids: Use graphs, charts, or tables to visually represent the data. This can help in making complex information more accessible and easier to understand.

6. PLOS authors have the option to publish the peer review history of their article (what does this mean? ). If published, this will include your full peer review and any attached files.

**Do you want your identity to be public for this peer review?** For information about this choice, including consent withdrawal, please see our Privacy Policy .

Reviewer #1: **Yes: ** Kausik Chatterjee

Reviewer #2: No

- - - - -

---

## [Author Response · Author response to Decision Letter 1]

23 Sep 2024

Reviewer 1

The paper investigates how physical activity (PA) varies between stroke survivors with differing ambulatory capabilities and compares them with healthy controls. Using accelerometers, the study characterizes and contrasts daily PA patterns, frequencies, and intensities among limited community ambulators, community ambulators, and healthy controls. It has several strengths:

• The paper addresses an important gap in the literature by exploring the PA patterns in post-stroke individuals with varying ambulation statuses, providing a nuanced understanding of their activity profiles.

• The use of accelerometers (Actigraph GT3X+) for objective measurement of PA provides accurate and reliable data, enhancing the credibility of the findings.

• The paper provides clear and practical recommendations for improving cardiovascular health in stroke survivors, emphasizing tailored interventions based on ambulation status.

• Ethical considerations were well-addressed, including approval from the Regional Board of Ethics in Stockholm and informed consent from participants.

However, it has a few major weaknesses:

Reviewer comment 1: Small Sample Size: The study only includes 39 participants post-stroke, limiting the generalisability of the findings to a broader population of stroke survivors.

Authors response: We appreciate the reviewer for raising this concern. While we acknowledge that the sample size is relatively small, which indeed limits the generalizability of the findings—a point we have already highlighted as a study limitation in the manuscript—we believe that this study offers valuable insights into the physical activity behavior of individuals post-stroke with varying ambulation statuses. This is particularly crucial for limited community ambulators, who are especially vulnerable to secondary health complications resulting from physical inactivity.

Reviewer comment 2: Limited Diversity in Participants: The study's participants are predominantly younger and have fewer women than the typical stroke population in Sweden, potentially affecting the applicability of the findings.

Authors response: We appreciate the reviewer’s comment regarding the limited diversity among the study participants. We acknowledge that the demographic imbalance is a limitation, and this has been noted in our manuscript. Despite this limitation, as mentioned in our response to Comment 1, we believe the study provides valuable insights into the physical activity patterns across different times of the day in people post-stroke with varying ambulation statuses. While the demographics may not fully represent the broader stroke population in Sweden, the findings still offer critical information that can inform interventions for specific subgroups, such as younger stroke survivors with different ambulation statuses. We agree that future studies should aim for a more diverse study sample to enhance the generalizability of the results, and we have recommended this approach for future research in the manuscript (Page 15, lines 281-284).

Reviewer comment 3: MVPA Cutoff Limitations: The MVPA cutoff used was developed for healthy older adults and might underestimate the intensity of PA for those with more severe disabilities. This could lead to misclassification of activity intensity, especially for the LCA group.

Authors response: We agree that the MVPA cut-points used in our study, which were originally developed for healthy older adults, may indeed underestimate physical activity intensity for individuals with more severe disabilities post stroke, such as limited community ambulators. Ideally, cut-points specific to people with stroke—and tailored to different subgroups of high and low-functioning individuals post-stroke—would be available and used. However, since these specific cut-points do not yet exist, we used the established MVPA thresholds to ensure consistency with existing literature and facilitate comparability across studies.

This limitation was acknowledged in our manuscript. Despite these constraints, we believe that the use of the ActiGraph accelerometer remains justified, as it is recommended in a recent consensus paper (Fini et al., 2023).

We have also highlighted the need for the development and validation of physical activity intensity cut-points that more accurately reflect the physiological capabilities of individuals with different ambulation statuses post-stroke, and have included this as a recommendation for future research (Page 15, lines 290-293).

How should we measure physical activity after stroke? An international consensus

Fini N, Simpson D, Moore S, Mahendran N, Eng J, Borschmann K, Moulaee Conradsson D, Chastin S, Churilov L, English C. Int J Stroke, 2023

Reviewer comment 4: Lack of Longitudinal Perspective: The study is cross-sectional, limiting the ability to infer causality or observe changes in PA patterns over time.

Authors response: We acknowledge that the cross-sectional design and the absence of longitudinal data are limitations of this study. However, these aspects are beyond the scope of our research aims, which focused on providing insights into physical activity behavior in individuals post-stroke with varying ambulation statuses compared to healthy controls.

Despite these limitations, this study is significant because, to our knowledge, it is the first to analyze physical activity patterns in relation to different bout durations among people with diverse ambulation impairments. This unique perspective fills a gap in the literature and offers valuable information that can inform future research and interventions.

In the discussion section of the manuscript (Page 16, lines 297-301), we now underscore the importance of conducting longitudinal studies to examine levels and patterns of physical activity post-stroke. This research is essential for addressing key questions, such as understanding the long-term effects of physical activity on stroke recovery and cardiovascular health.

Reviewer comment 5: Methodological Differences: The paper mentions differences in accelerometer placement and PA bout definitions compared to previous studies, which may affect comparability.

Authors response: The methodology used in our study was tailored to address our specific research question and was carefully considered during the study design. We believe that this approach was well-suited for accurately capturing physical activity patterns in individuals post-stroke with varying ambulation statuses.

We agree with the reviewer that this is a crucial methodological consideration. Therefore, we have included a discussion in the manuscript (page 16, lines 295-297) highlighting the need for future research to investigate how accelerometer placement affects the accuracy of capturing various domains of physical activity post stroke, such as step counts and moderate-to-vigorous physical activity (MVPA).

Reviewer comment 6: Data Sharing Restrictions: The data availability statement indicates restrictions due to privacy concerns, limiting access for further validation and research.

Authors response: Since data can indirectly be traced back to the study participants, according to the Swedish and EU personal data sharing legislation, access can only be granted upon request from the Research Data Office at Karolinska Institute (es.ik@odr). Any sharing of data will be regulated via a data transfer and use agreement with the recipient and require ethical approval from the Regional Board of Ethics in Stockholm. This is the official practice of Karolinska Institutet that we are required to follow.

Reviewer comment 7: Lack of Severe Cases: The exclusion of individuals with severe ambulation impairments limits the study’s applicability to all stroke survivors.

Authors response: We acknowledge that the lack of individuals with severe ambulation impairments limits the generalizability of our findings to the broader stroke population. The inclusion of individuals with mild and moderate impairments was aligned with our research question. We have addressed this limitation in the discussion of our manuscript (page 15, lines 281-284) and recommended that future research should include a broader range of impairments to enhance the applicability of the findings.

In addition, a few minor comments

Reviewer comment 8: Language and Clarity: Some sections could benefit from clearer language to enhance readability.

Authors response: All authors have thoroughly reviewed the manuscript and made revisions to improve readability where appropriate. Please let us know if there are any additional language revisions needed.

Reviewer comment 9: Figures and Tables: Ensure all figures and tables are clearly labeled and referenced in the text for better comprehension.

Authors response: Thank you for this suggestion. We have clarified the references to Table 1 in the results section (page 8, line 180).

Reviewer comment 10: Data Presentation: More detailed presentation of the data analysis methods could improve transparency and reproducibility.

Authors response: We thank the reviewer for this comment; however, we believe we have provided sufficient information to ensure the study’s transparency and reproducibility. If the reviewer feels otherwise, we would appreciate more specific feedback on how we can improve the data presentation.

Overall, this paper is more suited for the audience in the specialist physiotherapy journal rather than PLOS ONE.

Reviewer 2

The results presented in the current report require a more thorough and detailed analysis to fully understand their implications. It is essential to provide a comprehensive explanation of what the values signify in the context of the study. This includes interpreting the numerical data and explaining how these values relate to the objectives and hypotheses of the research.

A detailed breakdown should include:

Reviewer comment 1: Contextual Interpretation: Elaborate on how the values fit within the broader context of the study. This means discussing the relevance of these results to the research questions and how they align with or diverge from expected outcomes.

Authors response: We are unclear about the intent of this comment. Additional clarification would be helpful for us to address it effectively.

Reviewer comment 2: Statistical Significance: Clarify the statistical significance of the results. Provide information on the confidence intervals, p-values, or other statistical metrics used to determine the reliability of the findings.

Authors response: Thank you for your comment. We report categorical data as counts and percentages, and use medians and interquartile ranges for continuous data. Group comparison was performed using chi-square, Fisher’s exact test for categorical data, whereas Mann–Whitney U-test and Kruskal-Wallis test was used for continuous data. We believe that this approach to reporting and statistical analysis is appropriate to the nature and distribution of the data.

Reviewer comment 3: Comparative Analysis: Compare the results with previous studies or benchmarks. Explain how these values contribute to or challenge existing knowledge in the field.

Authors response: Thank you for this comment. We do feel that we have compared our results with previous studies in this field sufficiently throughout the discussion section. Please let us know if there were specific aspects of the results that would benefit from a more detailed comparison with existing literature.

Reviewer comment 4: Practical Implications: Discuss the practical implications of the results. What do these values mean for practitioners, stakeholders, or policymakers? How can they be applied in real-world scenarios?

Authors response: The paper discusses the impact of this work on both the scientific community and clinical practice, emphasizing the importance of tailoring physical activity promotion strategies based on an individual's ambulation status. Practically, this means that strategies for encouraging physical activity should be customized to fit each person's level of mobility. Furthermore, the paper highlights the need for additional research to specifically address physical inactivity in individuals with limited ambulation following a stroke.

Reviewer comment 5: Visual Aids: Use graphs, charts, or tables to visually represent the data. This can help in making complex information more accessible and easier to understand.

Authors response: We have included two tables (Table I: Study characteristics, Table II: Physical activity bouts) and 1 figure (Figure 1: Daily number of physical activity bouts during the morning, afternoon and evening). Please let us know if there are specific aspects of the results that could benefit from additional visual representations or tables.

---

## [Decision Letter · Decision Letter 1]

21 Mar 2025

PONE-D-24-27556R1Characterizing physical activity bouts in people with stroke with different ambulation statusesPLOS ONE

Dear Dr. Conradsson,

Thank you for submitting your manuscript to PLOS ONE. After careful consideration, we feel that it has merit but does not fully meet PLOS ONE’s publication criteria as it currently stands. Therefore, we invite you to submit a revised version of the manuscript that addresses the points raised during the review process. Please note that we have only been able to secure a single reviewer to assess your manuscript. We are issuing a decision on your manuscript at this point to prevent further delays in the evaluation of your manuscript. Please be aware that the editor who handles your revised manuscript might find it necessary to invite additional reviewers to assess this work once the revised manuscript is submitted. However, we will aim to proceed on the basis of this single review if possible.  The reviewer has requested a range of clarifications regarding the reporting of categorization, assessment and scoring tools used, among other matters, as well as some additional discussion points. Please ensure you address each of the reviewer's comments when revising your manuscript.

We look forward to receiving your revised manuscript.

Kind regards,

Hugh Cowley

Staff Editor

PLOS ONE

“This study was supported by the Norrbacka-Eugenia foundation, Promobilia foundation, and NEURO Sweden.”

3. In the online submission form, you indicated that [Since data can indirectly be traced back to the study participants, according to the Swedish and EU personal data sharing legislation, access can only be granted upon request from the Research Data Office at Karolinska Institute (rdo@ki.se). Any sharing of data will be regulated via a data transfer and use agreement with the recipient and require ethical approval from the Regional Board of Ethics in Stockholm.].

Additional Editor Comments (if provided):

Reviewers' comments:

Reviewer's Responses to Questions

**Comments to the Author**

1. If the authors have adequately addressed your comments raised in a previous round of review and you feel that this manuscript is now acceptable for publication, you may indicate that here to bypass the “Comments to the Author” section, enter your conflict of interest statement in the “Confidential to Editor” section, and submit your "Accept" recommendation.

Reviewer #3: (No Response)

2. Is the manuscript technically sound, and do the data support the conclusions?

Reviewer #3: Partly

3. Has the statistical analysis been performed appropriately and rigorously? 

Reviewer #3: I Don't Know

4. Have the authors made all data underlying the findings in their manuscript fully available?

Reviewer #3: No

5. Is the manuscript presented in an intelligible fashion and written in standard English?

Reviewer #3: Yes

6. Review Comments to the Author

Reviewer #3: Firstly, I am grateful for the opportunity to review the manuscript entitled: Characterizing physical activity bouts in people with stroke with different ambulation statuses which aimed to characterize and contrast the daily patterns, frequency and intensity of physical activity bouts between people post stroke with different ambulation statuses compared to healthy controls.

Here are some aspects to consider in relation to your manuscript in order to increase the quality of your manuscript.

Abstract.

In relation to the content of the abstract, it is important that the acronyms that appear are identifiable and that you have previously indicated what they refer to. It is especially suggested to pay attention to the acronyms LCA, CA, HC, MVPA, LIPA, which have not been previously indicated.

Such indication should also be considered in the manuscript.

Introduction.

It is suggested that the categorisation of the sample should be better defined, as the way it is expressed in the document may lead to confusion. In the case of the ‘sedentary exercisers’ group, it would be those who exercise 0.7 hours/day or more. Where are the subjects who exercise between 0.7 hours/day and < 0.5 hours/day included?

In the case of the ‘sedentary movers’ group, does it correspond to those subjects who do <0.5hours/day and > 0.1 hours/day?

In the case of the ‘sedentary prolongers’ group, are they those who perform = or < 0.1 hours/day?

It is really important that the categorisation of the sample is clearly stated in order to understand the results and the subsequent analysis and interpretation that follows.

Study participants.

In relation to the exclusion criteria, what assessment and scoring tools have been taken into account for the aspects of ‘cognitive impairments’, ‘severe neglect’ and ‘global aphasia’? It is recommended to provide such data to support the exclusion of participants.

Clinical assessments and performance-based clinical test

It is suggested that the authors provide the psychometric properties of the assessment tools used, as well as their validation and previous use in stroke survivors.

Assessment of physical activity.

It would be appreciated if the authors could provide a justification for the full 7-day monitoring with Actigraph GT3X+.

Table 1.

It is suggested that the authors include the data in relation to the group of ‘healthy controls’.

Discussion.

It would be appreciated to know why the use or non-use of assistive products has not been considered. On the other hand, it is suggested to provide such information in table 1.

It is recommended to review the use of the term ‘basic activities of daily living’, it is mentioned 2 times in the discussion section. If it is linked to ‘walking’ it is important to reflect to which type of mobility it refers as well as which of them involve energy expenditure.

Please provide information about the sample in relation to the performance of these basic activities.

References.

It is suggested that the authors update the references and include the most recent research and publications in both the introduction and the discussion.

7. PLOS authors have the option to publish the peer review history of their article (what does this mean? ). If published, this will include your full peer review and any attached files.

**Do you want your identity to be public for this peer review?** For information about this choice, including consent withdrawal, please see our Privacy Policy .

Reviewer #3: No

---

## [Author Response · Author response to Decision Letter 2]

4 Apr 2025

Reviewer 3

Firstly, I am grateful for the opportunity to review the manuscript entitled: Characterizing physical activity bouts in people with stroke with different ambulation statuses which aimed to characterize and contrast the daily patterns, frequency and intensity of physical activity bouts between people post stroke with different ambulation statuses compared to healthy controls.

Here are some aspects to consider in relation to your manuscript in order to increase the quality of your manuscript.

Comment 1 Abstract.

In relation to the content of the abstract, it is important that the acronyms that appear are identifiable and that you have previously indicated what they refer to. It is especially suggested to pay attention to the acronyms LCA, CA, HC, MVPA, LIPA, which have not been previously indicated.

Such indication should also be considered in the manuscript.

Author response: Thank you for bringing this concern to our attention. We acknowledge that the abstract and manuscript contain numerous abbreviations, which require clarification.

In the abstract, we have now clarified the abbreviations used for the two sub-groups of participants post-stroke:

- Limited community ambulators (LCA)

- Community ambulators (CA)

Additionally, to enhance clarity and minimize the number of abbreviations, we have decided to avoid using abbreviations for healthy controls, moderate-to-vigorous physical activity, and light physical activity in the abstract.

In the manuscript, we have now reviewed the use of abbreviations and made the following clarifications:

- We have now introduced CA and LCA in the second paragraph of the introduction (lines 55– 63).

- We have added definitions for LIPA and MVPA the first time these abbreviations appear in the introduction (lines 65-69).

- We have decided not to use the abbreviation for healthy controls (HC) to reduce the overall number of abbreviations.

Comment 2. Introduction.

It is suggested that the categorisation of the sample should be better defined, as the way it is expressed in the document may lead to confusion. In the case of the ‘sedentary exercisers’ group, it would be those who exercise 0.7 hours/day or more. Where are the subjects who exercise between 0.7 hours/day and < 0.5 hours/day included? In the case of the ‘sedentary movers’ group, does it correspond to those subjects who do <0.5hours/day and > 0.1 hours/day?

In the case of the ‘sedentary prolongers’ group, are they those who perform = or < 0.1 hours/day? It is really important that the categorisation of the sample is clearly stated in order to understand the results and the subsequent analysis and interpretation that follows.

Author response: Thank you for your comment. The categorization the reviewer is referring to comes from a separate referenced study and is not used as a cutoff to classify the present study sample. Instead, we categorized individuals post-stroke based on their self-selected walking speed: limited community ambulators (LCA) (self-selected gait speed <0.8 m/s) and community ambulators (CA) (self-selected gait speed ≥0.8 m/s) (see method section lines 142–146).

To clarify this, we have introduced CA and LCA in the second paragraph of the introduction (lines 55–63) and removed the reference to the study mentioned by the reviewer to avoid confusion. Furthermore, in this paragraph, we have emphasized the relationship between mobility status, gait speed, and physical activity in individuals post-stroke.

To support this, we have added the following references:

- Fini NA, Bernhardt J, Holland AEJD, rehabilitation. Low gait speed is associated with low physical activity and high sedentary time following stroke. 2021;43(14):2001-8.

- Bansal K, Fox EJ, Clark D, Fulk G, Rose DK. Speed- and Endurance-Based Classifications of Community Ambulation Post-Stroke Revisited: The Importance of Location in Walking Performance Measurement. Neurorehabil Neural Repair. 2024;38(8):582-94.

Additionally, in the light of the revisions and clarifications made in the second paragraph regarding categorization of ambulation statuses, we have made minor revisions to the text in the fourth paragraph of the introduction where the CA and LCA are mentioned lines 79-80). Finally, we have revised the aims to emphasize the categorization of the study sample (lines 92–94).

Comment 3. Study participants.

In relation to the exclusion criteria, what assessment and scoring tools have been taken into account for the aspects of ‘cognitive impairments’, ‘severe neglect’ and ‘global aphasia’? It is recommended to provide such data to support the exclusion of participants.

Clinical assessments and performance-based clinical test

It is suggested that the authors provide the psychometric properties of the assessment tools used, as well as their validation and previous use in stroke survivors.

Author response: We have now clarified (lines 107-110) that we used Montreal Cognitive Assessment (MoCA) to screen for cognitive impairments, Star Cancellation Test to screen for severe neglect and the ability to communicate through speech was evaluated informally during a telephone interview and while completing the MoCA. However, we did not use specific cut-off points for screening for e.g. cognitive impairments. Instead, we used these instruments to guide the screening process and make an informed decision about whether the person could participate in the test procedure in a safe and ethical manner, e.g., follow instructions during data collection.

Regarding the psychometric properties of the assessment tools used, we have now added references validating the performance-based clinical test for individuals post-stroke. The following references have been added to the manuscript:

Chedoke-McMaster Stroke Assessment (lines 125-126):

- Gowland C, Stratford P, Ward M, Moreland J, Torresin W, Van Hullenaar S, et al. Measuring physical impairment and disability with the Chedoke-McMaster Stroke Assessment. Stroke. 1993;24(1):58-63

Fatigue Severity Scale (line 129):

- Ozyemisci-Taskiran O, Batur EB, Yuksel S, Cengiz M, Karatas GK. Validity and reliability of fatigue severity scale in stroke. Top Stroke Rehabil. 2019;26(2):122-7.

Hospital Anxiety and Depression Scale (lines 133-135):

- Karlsson J, Hammarstrom E, Fogelkvist M, Lundqvist LO. Psychometric characteristics of the Hospital Anxiety and Depression Scale in stroke survivors of working age before and after inpatient rehabilitation. PLoS One. 2024;19(8):e0306754.

Montreal Cognitive Assessment (lines 137-140):

- Lau HY, Lin YH, Lin KC, Li YC, Yao G, Lin CY, et al. Reliability of the Montreal Cognitive Assessment in people with stroke. Int J Rehabil Res. 2024;47(1):46-51.

Comment 4 Assessment of physical activity.

It would be appreciated if the authors could provide a justification for the full 7-day monitoring with Actigraph GT3X+.

Author response: Thank you for your comment. We have now added a justification for measuring physical activity over seven days in the manuscript (see lines 152–154).

We have also added a reference to support this decision:

- Matthews CE, Hagstromer M, Pober DM, Bowles HR. Best practices for using physical activity monitors in population-based research. Med Sci Sports Exerc. 2012;44(1 Suppl 1):S68-76.

Comment 5. Table 1.

It is suggested that the authors include the data in relation to the group of ‘healthy controls’.

Author response: Thank you for this suggestion. In our manuscript, we have described the key characteristics of the healthy control group, specifically age and sex. Given that these are the only available variables for this group, we have opted not to present them separately in a table to maintain focus on the primary analyses.

Comment 6.

Discussion.

It would be appreciated to know why the use or non-use of assistive products has not been considered. On the other hand, it is suggested to provide such information in table 1.

Author response: Thank you for the suggestion. We have now included information on the use of walking aids in Table 1. It was more common to use a walking aid among limited community ambulators (82%) than in the community ambulators (41%) subgroup, which is now also highlighted in the discussion (lines 265–266).

Comment 7. It is recommended to review the use of the term ‘basic activities of daily living’, it is mentioned 2 times in the discussion section. If it is linked to ‘walking’ it is important to reflect to which type of mobility it refers as well as which of them involve energy expenditure.

Please provide information about the sample in relation to the performance of these basic activities.

Author response: We have reviewed the use of the term "basic activities of daily living" and have instead adopted the expression "low-intensity activities of everyday living", providing examples such as slow walking, household tasks, and gardening. These activities are commonly categorized as light-intensity physical activity (LIPA) but may represent moderate-to-vigorous physical activity (MVPA) for individuals with greater functional impairment. This distinction is now elaborated on in the discussion (lines 266–269).

Comment 8. References.

It is suggested that the authors update the references and include the most recent research and publications in both the introduction and the discussion.

Author response: Thank you for this suggestion. We have updated the references across the manuscript and made the following revisions:

In the introduction, we refer to the preventive role of regular physical activity in reducing the risk of stroke-related disability and stroke recurrence (Hobeanu et al., 2022, lines 43–45). Additionally, we have added a recent study by Espernberger et al. (2025) on the diurnal patterns of physical activity post-stroke (lines 76–77). We have also added two recent references regarding the link between ambulation status and physical activity post-stroke (see our response to comment 2 above).

In the discussion, we have added a recent studies by Hall et al (2023) on the interventions for behaviour change and self-management of risk in stroke secondary prevention (line 284).

- Hobeanu C et al. Risk of subsequent disabling or fatal stroke in patients with transient ischaemic attack or minor ischaemic stroke: an international, prospective cohort study. Lancet Neurol 2022

- Espernberger K et al. Physical activity patterns in independently mobile adult stroke survivors: an in-depth exploratory, observational study. Disabil Rehabil. 2025

- Hall P et al. Interventions for behaviour change and self-management of risk in stroke secondary prevention: an overview of reviews. Cerebrovasc Dis 2023

---

## [Decision Letter · Decision Letter 2]

9 Apr 2025

Characterizing physical activity bouts in people with stroke with different ambulation statuses

PONE-D-24-27556R2

Dear Dr. Conradsson,

We’re pleased to inform you that your manuscript has been judged scientifically suitable for publication and will be formally accepted for publication once it meets all outstanding technical requirements.

Kind regards,

Emiliano Cè, Ph.D.

Academic Editor

PLOS ONE

Additional Editor Comments (optional):

Reviewers' comments:

Reviewer's Responses to Questions

**Comments to the Author**

1. If the authors have adequately addressed your comments raised in a previous round of review and you feel that this manuscript is now acceptable for publication, you may indicate that here to bypass the “Comments to the Author” section, enter your conflict of interest statement in the “Confidential to Editor” section, and submit your "Accept" recommendation.

Reviewer #3: All comments have been addressed

2. Is the manuscript technically sound, and do the data support the conclusions?

Reviewer #3: Yes

3. Has the statistical analysis been performed appropriately and rigorously? 

Reviewer #3: Yes

4. Have the authors made all data underlying the findings in their manuscript fully available?

Reviewer #3: No

5. Is the manuscript presented in an intelligible fashion and written in standard English?

Reviewer #3: Yes

6. Review Comments to the Author

Reviewer #3: The authors have provided clarifications on the comments made and included the necessary changes to the manuscript in an appropriate manner.

7. PLOS authors have the option to publish the peer review history of their article (what does this mean? ). If published, this will include your full peer review and any attached files.

**Do you want your identity to be public for this peer review?** For information about this choice, including consent withdrawal, please see our Privacy Policy .

Reviewer #3: No

---

## [Editor Report · Acceptance letter]

PONE-D-24-27556R2

PLOS ONE

Dear Dr. Moulaee Conradsson,

I'm pleased to inform you that your manuscript has been deemed suitable for publication in PLOS ONE. Congratulations! Your manuscript is now being handed over to our production team.

Kind regards,

on behalf of

Prof. Emiliano Cè

Academic Editor

PLOS ONE